# ScaleCert: Scalable Certified Defense against Adversarial Patches with Sparse Superficial Layers

**Husheng Han**[1,2,3] *    **Kaidi Xu**[4] *    **Xing Hu**[1] †    **Xiaobing Chen**[1,2,3]    **Ling Liang**[5]
**Zidong Du**[1,3]    **Qi Guo**[1]    **Yanzhi Wang**[6]    **Yunji Chen**[1,2]

[1]SKL of Computer Architecture, Institute of Computing Technology, CAS
[2]University of Chinese Academy of Sciences  [3]Cambricon Technologies
[4]Drexel University  [5]UC Santa Barbara  [6]Northeastern University
{hanhusheng20z,huxing,chenxiaobing,duzidong,guoqi,cyj}@ict.ac.cn
kx46@drexel.edu, yanz.wang@northeastern.edu liangling@cs.ucsb.edu

## Abstract

Adversarial patch attacks that craft the pixels in a confined region of the input images show their powerful attack effectiveness in physical environments even with noises or deformations. Existing certified defenses towards adversarial patch attacks work well on small images like MNIST and CIFAR-10 datasets, but achieve very poor certified accuracy on higher-resolution images like ImageNet. It is urgent to design both robust and effective defenses against such a practical and harmful attack in industry-level larger images. In this work, we propose the certified defense methodology that achieves high provable robustness for high-resolution images and largely improves the practicality for real adoption of the certified defense. The basic insight of our work is that the adversarial patch intends to leverage localized superficial important neurons (SIN) to manipulate the prediction results. Hence, we leverage the SIN-based DNN compression techniques to significantly improve the certified accuracy, by reducing the adversarial region searching overhead and filtering the prediction noises. Our experimental results show that the certified accuracy is increased from 36.3% (the state-of-the-art certified detection) to 60.4% on the ImageNet dataset, largely pushing the certified defenses for practical use.

## 1   Introduction

Despite the promising opportunities and great success of deep neural network (DNN) techniques in computer vision tasks [9, 13], DNN techniques are vulnerable to adversarial attacks that craft the input data with adversarial perturbations to manipulate the output [2, 20, 26]. Recent studies reach a consensus that compared to the adversarial example attack, adversarial patch attack is more practical and well-demonstrated in physical environment with good effectiveness and transferability in both classification and detection systems [5, 14, 15, 23, 25, 28]. The key difference between the adversarial example attack and patch attack is the constraint of perturbation. For adversarial example attack, the adversary perturbs all pixels in the image with the $\ell_p$-norm of the perturbation in prescribed bounds, while the adversarial patch attack only changes the pixels in a confined region without bounded constraints but free to choose the values. Consider the real-world condition, it's hard to define the $\ell_p$-norm constraint of the perturbation due to the environment, viewpoint and object variation. Hence, the adversarial patch attack is much more appropriate to be mounted in the physical world. The success of practical adversarial patch attacks raises the importance and urgency of defense methodologies. Existing defending methodologies can be classified into two categories, heuristic defenses and certified defenses. For heuristic defense approaches, they have good efficiency

---

∗ Equal contribution. † Corresponding author.

but lack robustness against a strong adaptive attack. For example, digital watermarking (DW) [8] utilizes the magnitude of the saliency maps to detect unusually dense regions and mask them out of the input. Local gradient smoothing (LGS) [19] pre-processes the image gradient of the classification function with a normalization and a thresholding step and then suppress the adversarial noise based on the gradient. However, these empirical defenses may be invalidated when confront with the strong adaptive attacker that has the white-box knowledge of the defense [3]. For certified defenses, they must be proved that have provable robustness, like [3] proposes the first certified defense based on the interval bound propagation (IBP) [7] that are commonly used in adversarial robustness certification problems. De-randomized smoothing (DS) [15] proposes the defense method extending randomized smoothing robustness schemes [4] with structured ablation. Moreover, a provable robustness defense based on clipped BagNet [1] with small reception field is proposed by [33] . However, all of these certified studies achieve distinctly low certified accuracy for large scale datasets such as ImageNet [13]. A certified accuracy of 20.5%-36.3% [1] from these methods can hardly be applied in the real world systems. It is a dilemma that certified robustness against such a practical and pernicious attack model can be hardly scaled for realistic usage.

To address this issue, in this paper, we propose the defenses with both certified robustness and good scalability for large scale dataset that used in practical scenarios. Inspired by the existing analysis on the unique distribution of the activation map yielded by adversarial patch attacked input [16, 19, 25], and substantial neural network compression techniques without sacrificing natural accuracy [10, 31], we observe that the adversarial patches rely on the localized superficial important neurons (SINs) to poison the output and can be exceedingly alleviated by utilizing the pruning techniques. We then propose the scalable certified (ScaleCert) defense with SIN-based neural network sparsity approach. The results show that the certified accuracy of ScaleCert on ImageNet dataset achieves 60.4% against 1%-pixel patch, significantly surpassing the highest record of 36.3% in state-of-the-art certified detection methods [24]. This is the first certified work that largely reduces the gap between clean and certified accuracy, retaining the high certified accuracy with good computing efficiency, which largely pushes the certified defenses for common practice in real adoption.

## 2 Problem Setup

In this section, we describe the formulation and important terminology of adversarial patch attack and certified defenses against it.

### 2.1 Adversarial Patch Attack Formulation

Given an input $\mathbf{x} \in \mathcal{X} \subset [0,1]^{W \times H \times 3}$ with a true label $y^*$, adversarial patch attack can manipulate the output label of victim model, $\mathcal{M}(\cdot)$, by adding the patch perturbation on $\mathbf{x}$. The goal of adversarial patch attack is to find an adversarial patch $\hat{\mathbf{x}}$ that can be added on $\mathbf{x}$ to generate image $\mathbf{x}' \in \mathcal{A}(\mathbf{x})$ satisfying a constraint $\mathcal{A}$ such that $\mathcal{M}(\mathbf{x}') = y' \neq y^*$. $y'$ can be a targeted label (targeted attack) or any arbitrary label not equal to the benign label $y^*$ (untargeted attack). The adversarial patch $\hat{\mathbf{x}}$ can be generated by solving the follow equation:

$$\hat{\mathbf{x}} = arg \max_{\mathbf{x}' \in \mathcal{A}(\mathbf{x})} \mathbb{E}_{\mathcal{X}}[logPr(\mathcal{M}(\mathbf{x}') = y'|\mathbf{x})] \tag{1}$$

The constrain $\mathcal{A}(\mathbf{x})$ is determined by the threat model. In this paper, the adversary is allowed to arbitrarily change pixels within a square contiguous region and locate this region *anywhere* on the image, which is similar to the previous works [15, 18, 25]. Formally, we use a binary pixel block $\mathbf{p} \in \{0,1\}^{W \times H}$ to represent the restricted square region, where the pixels located in the region are set to one, otherwise zero. For the purpose of stealthiness, $\mathbf{p}$ is extremely smaller than $\mathbf{x}$, usually has the size of 1% to 5% of the $\mathbf{x}$. The constraint set $\mathcal{A}(\mathbf{x})$ can be expressed as:

$$\mathcal{A}(\mathbf{x}) \in \{\mathbf{x}' = (\mathbf{1} - \mathbf{p}) \odot \mathbf{x} + \mathbf{p} \odot \hat{\mathbf{x}}\} \tag{2}$$

where $\odot$ refers to the element-wise product operator, and $\hat{\mathbf{x}}$ is the adversarial patch we generated from Eq. (3).

---

[1]Covering both certified recovery and detection methodologies. See Table 2 for more details.

## 2.2 Certified Defenses

Bounding the range of a neural network outputs given a $\ell_p$-norm of input perturbation has become an important theme for neural network verification and certified adversarial defense [4, 7, 21, 22, 27, 32]. Recently, researchers also shed light on generalizing the conventional input perturbation constraint to the adversarial patch formulation. Existing certified defenses against adversarial patch attacks can be classified into two major categories with different certification problems:

1) **Certified defense for attack recovery.** The classifier returns the assurance that the classification result will provably not change under any distortion within an adversarial constraint set for the input. It is a heritage from certified defenses towards adversarial example attacks. Formally, the classifier ensures $\mathcal{M}(\mathbf{x}_0) = \mathcal{M}(\mathbf{x}') = y_0$ to produce a certificate for an clean input $\mathbf{x}_0$ for any adversarial example $\mathbf{x}' \in \mathcal{A}(\mathbf{x}_0)$, where $\mathcal{A}(\mathbf{x}_0)$ is the adversarial constraint set defined in Eq.( 2). Many existing studies extend the certified techniques in adversarial example attacks with the new constraints of patch perturbation to propose the certified defense against patch attacks. For example, [3] proposes their certified defense by extending interval bound propagation (IBP) [7] defense on patch attacks. [15] proposes the derandomized smoothing methodology based on structured ablation scheme by extending the randomized smoothing technique [4]. PatchGuard [25] leverages the DNN models with small reception fields to localize the adversarial features and proposes the robust masking defense to remove adversarial effect of these features.

2) **Certified defense for attack detection.** The classifier returns the assurance that the image is clean or alert when the image is an adversarial input. It is a relaxed certified problem compared to certified recovery. However, it is also an important and effective defense because occluding the adversarial patches can eliminate the adversarial effect. MRD [18] is the first work that proposes the certified attack detection study. By examining the pattern of the prediction labels that occlude the windows of the images, the adversarial images can be certified detected. PatchGuard++ [24] aims to improve the certified detection rate by extending PatchGuard with the same insights of MRD.

For both certified recovery and detection defenses, the state-of-the-art approaches obtain low certified accuracy on large scale datasets and can be hardly adopted for practical use. As shown in Table 1, the best certified accuracy in recent literature is about 36.3% when the patch contains 1% pixels of the input image. Therefore, we confront with the dilemma of practicality and low certified provable robustness for the harmful and practical adversarial patch attacks. To address this issue, we aim to propose a defense methodology that bridges the gap between clean and certified accuracy and achieves both provable robustness and good scalability for large scale datasets.

Table 1: Certified accuracy of state-of-the-art defenses on ImageNet.

| Certified Defense | Certified Detection | | Certified Recovery | | |
|---|---|---|---|---|---|
| (ImageNet, PatchSize=1%) | PatchGuard++ [24] | PG-Mask-BN[25] | PG-Mask-DS [25] | CBN [1] | DS [15] |
| Certified Accuracy | 36.30% | 32.30% | 22.50% | 21.90% | 20.50% |

## 3 ScaleCert Defense

In this section, we first use an empirical study to motivate the use of superficial important neurons (SIN) distribution to distinguish the adversarial patch and eliminate the adversarial effect and noises. Then we propose the ScaleCert to defend against adversarial patch attack with SIN-based sparsification methodology. To note, Section 3.1 focuses on empirical analysis based on the results of some specific adversarial patch attack methodology. However, the ScaleCert's robustness applies to any arbitrary adversarial patch attack. The provable robustness of this defense is demonstrated and analyzed in Section 3.3.

### 3.1 SIN in Patched and Benign Images

In observing that the adversarial patch determines the prediction results with a very small region of pixels for a relatively broad range of input images, we thus perform superficial feature importance distribution analysis of patched images and benign images. Though neuron importance analysis has been widely used for abnormal input detection [6, 16, 29, 30], the metric in our methodology (superficial feature importance) is distinct from previous studies (deep feature importance). The latter focuses on the neurons that contribute significantly to the inference output, while superficial important

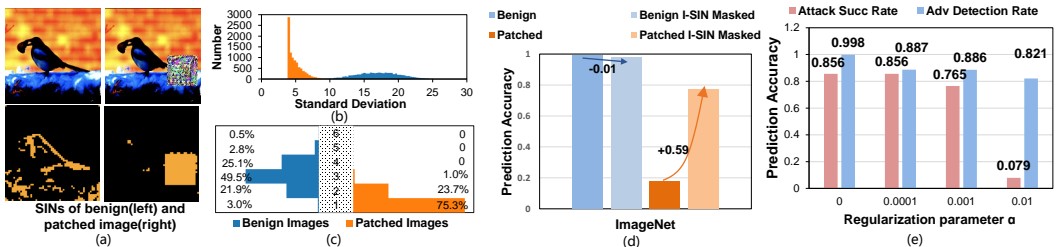

Figure 1: SIN analysis in benign and patched images.(a) A showcase of SINs in the benign and patched image. (b) The distance deviation of the SIN. (c) The average important neuron cluster number distribution. (d) The prediction stability of moving out localized SINs in benign and patched images. (e) Adaptive attack effectiveness by steering around the detection of SINs

neurons refers to the neurons that contribute significantly to the shallow feature map values in the first (several) layer(s). Compared to deep important neurons (DIN), SIN has good discrimination to distinguish patched images, more straightforward correlation with the input adversarial regions, and smaller computing overhead without requirement for gradient calculation. More comparison between SIN and DIN is in Appendix A.3.

Specifically, we discover that the SIN of patched images exhibits extremely localized pattern and the adversarial patch effects can be eliminated if localized SINs are removed. For benign images, however, the prediction results do not rely on the localized SINs and thus being more stable. ***The intuition** is that the prediction of typical DNN inference on the benign inputs depends on the deep features of the input image, while adversarial inputs have to strengthen the superficial features. Otherwise, its adversarial effect will fade out through the computation of all DNN layers.* We quantitatively evaluate the impact of SIN during predictions of benign and patched images as follows.

1) ***Localized SIN Feature Analysis***: *SINs in patched images are much more localized than that in benign images*. We show the Top-200 neurons in the activation map of the first layer for benign images and patched images in Figure 1(a). Then, we compare the standard deviation of distance between the Top-200 neurons to the central neuron of benign images and patched images for randomly-selected 10,000 images in ImageNet dataset. The patched images' distance deviation is much smaller than benign images, as shown in Figure 1(b). Additionally, we also cluster the Top-k neurons with the classic MeanShift clustering algorithm and observe the cluster number of SINs in patched images is much less than that in benign images. As shown in Figure 1(c), for patched images, about 75% of cases have only one cluster and 99% of cases have less than two clusters. While for benign images, more than 75% of the cases have more than three clusters. The results indicate that SINs are much more localized in patched images in terms of both the distance and cluster number.

2) ***Prediction Stability Analysis***: *By occluding the localized SINs in a small window, the prediction results of patched images are unstable, while benign images seldom change.* Specifically, we examine when moving the localized SIN features, how the prediction results are affected in patched images and benign images. We compare the prediction accuracy of benign objects and patched objects before and after moving out the localized SINs (l-SIN), as shown in Figure 1(d). In this experiment, we consider 10000 randomly-selected ImageNet images that can be correctly classified by the model to eliminate the interference of other factors. The results show that for benign images, the prediction accuracy is not sensitive to the localized SIN features. For the patched images, originally the prediction accuracy is extremely low because of the patch effects. After moving the localized important features, the detection rate increases drastically, which indicates that the patch effect has been effectively eliminated. The results show that the nature images perform robust classification without relying on extremely localized SIN features, while patches rely on the extremely localized SIN features to deceive and induce the object detector to output the incorrect results.

3) ***Adaptive Attack Against SIN-based Detection***: *The adversarial patch attacks confront with the dilemma of either introducing large SINs or hardly affecting the prediction results.* We perform the adaptive attacks on ImageNet dataset and the results are shown in Figure 1 (e). To steer clear of the detection based on SINs, the adversary trains the adversarial patch by taking the superficial activation value into consideration. Compared to Equation(1), a penalty loss of activation value is considered with the parameter $\alpha$, when $\alpha$ is larger, the attack pays more attention to the stealthiness other than the attack effectiveness. In this way, the adversary aims to build the adversarial patch with both good poisoning effects, but also trying to escape from the adversary candidate searching.

Adversary detection rate refers to the rate that detector correctly identifies the adversarial region in the adversarial inputs or the benign input with no adversarial region. The results show that, it is indeed that the adversary detection rate decrease to 82.1% when adversary attempts to reduce the activation magnitude in the first layer ($\alpha$ increases from 0 to 0.01). However, compared to the gentle slope of adversary detection rate, the attack success rate decreases much more drastically. When $\alpha$ is increased from 0 to 0.01, the adversarial attack success rate drops to 7.9%. These results indicate that the adversary cannot maintain the two goals of high attack success rate and good stealthiness in SINs simultaneously. The detailed attack setup and the datasets are covered in Appendix A.1.

These results show the empirical evidence to the hypothesis that the adversarial patch attacks rely on the strengths of localized SINs to effectively damage the prediction results of victim models. We leverage such insights to design ScaleCert certified defense methodology. Different from the motivation of previous studies [25], we don't constrain the reception field to localize the adversarial feature effect in the deep features, but focus on the elimination of poisoned SIN features. More comparison and discussion between these two design philosophies are provided in Appendix A.3.

### 3.2 ScaleCert Framework

**SIN:** Giving the feature map of the superficial important layer, we sort the neurons by the sum of their activation through channels and regard the top-k neurons as the SIN. SIN is not static but dynamic for each image during their inference. for occluding region Suppose we use the layer $s$, and the $\mathcal{N}_i^s$ is notated as the feature map of $i$-th channel of layer $s$. The SIN($n^s$) is defined as follows:

$$\mathrm{SIN}(n^s) = Top_k(\mathcal{N}^s) \quad where \quad \mathcal{N}^s = \sum_{i=1}^{c} \mathcal{N}_i^s$$

**SIN-mask**: SIN-mask is dynamically generated according to SINs when taking in different inputs. Specifically, SIN-mask is 2 dimensional and is the same size as the feature map of the superficial important layer. Giving the feature map of the superficial important layer, SIN-mask is the mask where the positions of SINs are set to 1 and others are set to 0.

**ScaleCert Overview.** ScaleCert consists of three stages: 1) *SIN-based Pruning*: Calculating the SIN-mask, $\mathcal{S}$, and pruning the rest of neurons . We first select the layer $s$ as the superficial important layer, then we keep Top-$k$ activation neurons only and prune other unimportant neurons of this layer, where $k$ is the top-ranking ratio. 2) *Occluding Prediction*: With the SIN-mask of the superficial layer, we calculate the candidate searching region $\mathcal{R}$ on the input image $x$ that contributes to the SIN. Then we impose sliding windows on $x$ and check their overlaps with the candidate searching region $\mathcal{R}$. ScaleCert makes inference by masking out every sliding window that $\mathbf{w} \cap \mathcal{R} \neq \emptyset$ with dynamic SIN-based pruning independently, and obtain the occluding prediction map. The size of sliding window $\mathbf{w}$ is $k \times k$, where $k$ equals to the sum of patch size ($p$) and an additional step ($r - 1$) ($r$ is 3 in the evaluation setting). Therefore, there are at least $r \times r$ windows that completely occlude the adversarial patch. 3) *ScaleCert Detection*: Finally, check the labels of the prediction map to detect whether there is an adversarial patch. If all the labels in the masked predictions are consistent, there is no attack and return the original model prediction $\mathcal{M}(\mathbf{x})$. ScaleCert leverages SIN-based sparsity to not only reduce the occluding windows, but more importantly filter out the noisy labels in trivial prediction maps and achieves the state-of-the-art certified accuracy.

**Neural Network Sparsification.** The key success factor of ScaleCert is leveraging the pruning technique on the superficial layer $s$ to improve certified accuracy and efficiency. In order to maintain the clean accuracy of our ScaleCert, we need to finetune the model with the sparse superficial layers. Figure 2 illustrates the the proposed winner-take-all dropout. We add the SIN-mask after the superficial layer(s) and prune the non-important neurons according to the activation magnitude. Only the top-ranking neurons (winners) can forward their outputs to the following layers. To finetune the sparsified neural networks, we only update the top-ranking neurons selected by the SIN-mask. This finetuning procedure can be treated as a shallow layers channel pruning, the SIN-mask, $\mathcal{S}$ will also be updated during each iteration.

**Occluding Prediction.** With the SIN mask $\mathcal{S}$, we first calculate the input region $\mathcal{R}$ that contributes to the SINs. The calculation procedure is as follows: for the neuron with the ordinate of $(x_s, y_s)$ in superficial neural network layer $s$, its antecedent neurons in layer $s - 1$ that contribute to the value of this neuron are in the square region with the top left ordinate $(x_s * s_s - p_s, y_s * s_s - p_s)$ and the bottom right ordinate $(x_s * s_s - p_s + k_s - 1, y_s * s_s - p_s + k_s - 1)$, respectively, where $p_s, s_s$

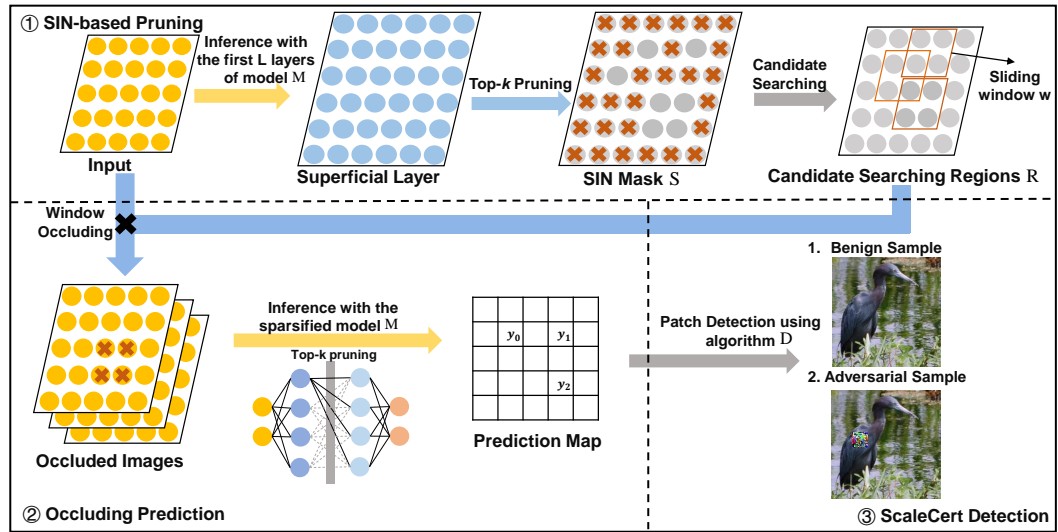

Figure 2: ScaleCert Framework with three stages: SIN-based Pruning; Occlude Prediction; and ScaleCert Detection.

and $k_s$ are the padding size, stride size and kernel size of layer $s$. Then, we can iteratively calculate region in the inference pyramid until back to the initial input layer and obtain region $\mathcal{R}$. $\mathcal{R}$ is the candidate region for occluding prediction, while all the pixels falling out of this region cannot affect the prediction results because of the SIN mask.

Then we impose the sliding window $\mathbf{w}$ across all the $\mathcal{R}$ region and obtain the occluding prediction map by moving out the pixels in the sliding window. Note that, we also mask the superficial features of occluding region by set their values as the **_lowest_** one in this feature map when masking the input pixels to make sure the occluded region locates out of the SIN area, therefore, not affecting the prediction. The number of candidate window $\mathbf{w}$ has been reduced because of SIN mask. To reduce the overhead of inference procedures in the further step, we additionally propose to merge adjacent occluding windows when the overlapped area among every two windows exceed the threshold ($\tau$). To note, the merging operations assure to cover every occluding window $\mathbf{w}$ that has overlap with $\mathcal{R}$ for the certified robustness.

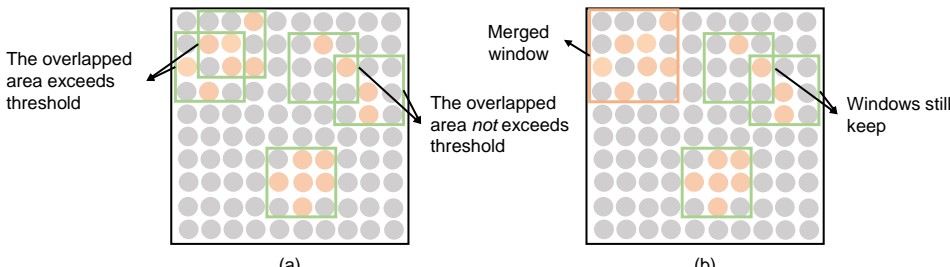

Figure 3: Merge sliding occluding windows. (a) Original windows. (b) Merged windows.

**Detection Methodology** The detailed ScaleCert detection algorithm is given in Algorithm 1. ScaleCert only imposes sliding window on the $\mathcal{R}$ based on the SIN area since other neurons will be masked out during inference. Interestingly, the SIN-mask not only helps on saving time but also filters out most noisy area on the original images so that makes the $y_i$ more consistent in SIN area and benefits on improving the clean accuracy on benign images. For patch attacked cases, there must be a $r \times r$ region that the label will be changed when we masking out this area. So the algorithm will alert (line 12) to user and provide the potential patch location that can be used to empirically recover true label by occluding the area. In some rarely happened cases, like there are noises on the input even in the SIN area, we cannot make the promise decision, and conservatively raise alarm (line 14).

**Algorithm 1** Our adversarial patch detection algorithm $\mathcal{D}$
___
1: **Inputs**: image $\mathbf{x}$, model $\mathcal{M}$, sliding windows on the input $\mathcal{W}$ and candidate searching region $\mathcal{R}$
2: **Output**: Prediction $\mathcal{M}(\mathbf{x})$ or alert
3: $\mathbb{S} \leftarrow \emptyset$
4: $y^* \leftarrow \mathcal{M}(\mathbf{x})$
5: **for** each $\mathbf{w}_i \in \mathcal{W}$ **do**
6:     **if** $\mathbf{w}_i \cap \mathcal{R} \neq \emptyset$ **then**                    ▷ Filter out some potential sliding windows
7:         $y_i \leftarrow \mathcal{M}(\mathbf{x} \odot (1 - \mathbf{w}_i))$      ▷ Obtain the label according to the mask-out input
8:         **if** $y_i \neq y^*$ **then**
9:             $\mathbb{S} \leftarrow (\mathbf{w}_i, y_i)$      ▷ Collect potential malicious windows and prediction labels
10: **if** $\mathbb{S} \neq \emptyset$ **then**
11:     **if** $\mathbb{S}$ contains $y'$ **and** $y_i = y'$ **for** $\forall \mathbf{w}_i \in \mathcal{R}_{r \times r}$ **then**
12:         **return** alert              ▷ Adversarial patch detected at $\mathcal{R}_{r \times r}$ area
13:     **else if** MAJORITYVOTING($y_i$) $\neq y^*$ **then**
14:         **return** alert              ▷ Possible adversarial patch with noises in $\mathcal{S}$
15:     **else**
16:         **return** $y^*$                     ▷ benign image with noises in $\mathcal{S}$
17: **else**
18:     **return** $y^*$                          ▷ benign image

---

**Algorithm 2** Our certify algorithm $\mathcal{C}$
___
1: **Inputs**: image $\mathbf{x}$, label $y$, model $\mathcal{M}$, sliding windows on the input $\mathcal{W}$ and candidate searching region $\mathcal{R}$
2: **Output**: Whether the image $\mathbf{x}$ has provable robustness to label $y$
3: **for** each $\mathbf{w}_i \in \mathcal{W}$ **do**
4:     **if** $\mathbf{w}_i \cap \mathcal{R} \neq \emptyset$ **then**
5:         $y_i \leftarrow \mathcal{M}(\mathbf{x} \odot (1 - \mathbf{w}_i))$
6:         **if** $y_i \neq y$ **then**
7:             **return** False
8: **return** True

---

### 3.3 Provable Robust Analysis

**ScaleCert Certification** In this paper, we assume the attacker has full knowledge of our defense method and is allowed to arbitrarily change pixels in a square region based on any needed information of the threat model. Our detection defense method ScaleCert aims to provably ensure that certified image can be correctly classified or yields an alarm together with potential patch location and maintain high accuracy on benign images.

**Theorem 1** *If Algorithm 2 returns True for a given image* $\mathbf{x}$*, our defense in Algorithm 1 can either make a correct prediction or alert on any adversarial image* $\mathbf{x}' \in \mathcal{A}(\mathbf{x})$

**Proof**: The Algorithm 2 returns True indicates that all single sliding window that $\mathbf{w}_i \cap \mathcal{R} \neq \emptyset$ do not affect the prediction results for image $\mathbf{x}$ like Figure 4(b) while the effect of other windows are all eliminated since the top-k pruning like Figure 4(c). Then if we apply adversarial patch attack on $\mathbf{x}$, which means any $\mathbf{x}' \in \mathcal{A}(\mathbf{x})$ can be fed to Algorithm 1, there are two possible cases:

- Case 1: The adversarial patch overlaps with the SIN area of certified benign image like Figure 4(d) and leads a malicious label. For occluding windows that not overlap with the patch, the outputs are not changed like Figure 4(e). However, consider of the size of our sliding window and the patch size are $k \times k$ and $p \times p$, and respect to the condition that $k = p + r - 1$, there must have $r \times r$ sliding windows that completely mask out the adversarial patch like Figure 4(f), and the new pruned SIN area is the same as the benign images with a same occluding window like Figure 4(b), so the masked out prediction is exactly the true label. Therefore, in this case Algorithm 1 returns alert (line 12).

- Case 2: The adversarial patch locates outside the SIN area like Figure 4(g) and outputs a malicious label. Occluded images that not occluding the patch still maintain the malicious

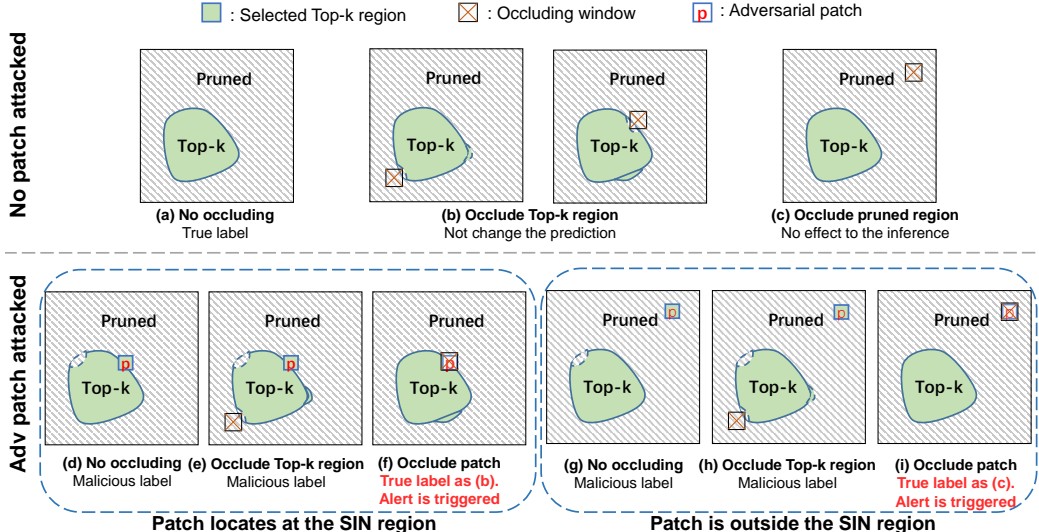

Figure 4: The certified benign image with only Top-k region retained(a) and occluded images that masking benign image out with occluding windows (b, c) are all classified as true label(Algorithm 2). Attacked images that patch locates at the Top-k region(d, e, f) or outside the Top-k region (g, h, i) will alert (f, i) that ensures the correctness of certification.

labels. But, there also exists $r \times r$ windows that masked out the patch completely and recover the correct label like Figure 4(i) which triggers the alarm (Algorithm 1 line 12). ☐

Note that, the line 13-16 in Algorithm 1 is not necessary to the certified defense, since the patch is provably caught in line 11 when the input x returns `True` in Algorithm 2. They are designed for the benign image with noises in SIN area or both patch and noises happened in SIN area. In summary, Algorithm 1 is a general empirical adversarial patch detection method but when combining with the Algorithm 2, it can conclusively output normal predictions on clean images but alert when an attack takes place for the certified inputs.

## 4 Experiments

### 4.1 Experiment Setup

**Evaluation Metrics** Certified accuracy: note that our Algorithm 1 can return a provable answer if the input image is benign or can be recovered by our algorithms, but it may be incapable of rare examples that violate our assumption. So the certified accuracy we defined here is a lower bound of the real certified accuracy.

**Datasets and Models.** We evaluate the clean and certified robustness results on the high resolution dataset with $224 \times 224$ sizes (1000-class ImageNet [13]) and low resolution dataset with $32 \times 32$ sizes (10-class CIFAR-10 [12]). In our evaluation, we use ResNet50 [9] for ImageNet dataset and ResNet18 [9] for CIFAR-10 dataset. We train the models with pretrained weights and finetune them with SIN-based pruning and window occluding of random locations to improve the performance of inference.

**Baselines.** We compare the defense performance with the state-of-the-art studies: PatchGuard++ [24], PG-mask-BN [25], PG-mask-DS [25], Clipped BagNet (CBN) [1], De-randomized Smoothing [15], and Interval Bound Propogation based certified defense (IBP) [3]. We use the optimal parameter settings in their configurations obtained from their reports.

**Attack Patch Size** We adopt the similar configurations from previous studies that defend against a single sequare adversairal patch that consists up to 1%, 2%, or 3% pixels of the ImageNet images, and 0.4% and 2.4% pixels of CIFAR-10 images. Additionally, we also report the extremely large patches with 5% and 8% in the Appendix A.4.

## 4.2 Provable Robustness Results

The certified accuracy results of ScaleCert and other recent studies are presented in Table 2. For ScaleCert, 10 trials have been run to get the average results with different random seeds. The small variance that 0.89 for clean accuracy and 0.27 for certified accuracy for ImageNet with a patch of 1% pixels indicates that ScaleCert can give stable results. For PatchGuard++, MRD, and ScaleCert, the certified accuracy refers to the certified detection accuracy. For others, the certified accuracy refers to the certified recovery accuracy. These results hold for any attack within the patch size constrain. We draw the following conclusions from the results.

*Existing studies achieve good certified accuracy in low-resolution images, but very poor provable robustness in high-resolution images.* For example, when patch size is 2.4% of the images, PatchGuard++, PG-Mask-DS, PG-Mask-BN achieves 74.1%, 57.7%, and 47.3% of certified accuracy. For ImageNet dataset, their certified accuracy decreases to 33.9%, 19.0% and 26.0%. There are two potential reasons behind this phenomenon: 1) Deeper models are used for the classification of high-resolution images. With deeper layers, the reception field becomes larger even using BagNet with limited small kernel size. It is hard to distinguish and eliminate the adversarial effect at the deep feature extraction layer. 2) For high-resolution images, the random ablation in DS-based algorithms is unable to produce satisfactory accuracy due to the loss of the most information of the input images.

*MRD and IBP are computational infeasible for high-resolution images.* MRD requires to compute the prediction map for every occluding window, which introduces infeasible computational overhead with $O(W \times H)$ inference rounds. For example, MRD requires about 36481 inference rounds when the patch is 2%, which costs about 49 $s$ for V100 GPU platform to verify a single input image with 224×224 pixels even after parallelization optimization. As a comparison, our study reduces the candidate windows number to less than 100, which significantly removes the computation overhead.

*ScaleCert outperforms other defenses with much higher certified accuracy for ImageNet datasets, which pushes the certified defenses into practical usage.* ScaleCert achieves both good certified accuracy for low-resolution images and high-resolution images. For high resolution images, particularly, the experimental results show that ScaleCert achieves the best certified accuracy. Specifically, when patch contains 1% pixels of the input images, ScaleCert achieves 60.4%, largely surpassing the PatchGuard++ with 36.3% of certified accuracy. CIFAR-10 images are very small with the only size of $32 \times 32$, and thus are much more sensitive to pruning operations and masking operations. Therefore the certified accuracy of ScaleCert drops slightly compared to MRD.

Table 2: Comparison of clean and certified accuracy under different defenses

| Dataset | | ImageNet | | | | | | Cifar | | | |
|---|---|---|---|---|---|---|---|---|---|---|---|
| Patch size | | 1% pixels | | 2% pixles | | 3% pixels | | 0.4% pixels | | 2.4% pixels | |
| Accuracy | | clean | robust | clean | robust | clean | robust | clean | robust | clean | robust |
| Certified Recovery | IBP | computationally infeasible | | | | | | 65.8 | 51.9 | 47.8 | 30.3 |
| | CBN | 53.5 | 21.9 | 53.5 | 13.7 | 53.5 | 7.4 | 83.2 | 51.0 | 83.2 | 16.2 |
| | DS | 44.4 | 20.5 | 44.4 | 17.0 | 44.4 | 14.9 | 83.9 | 68.9 | 83.9 | 56.3 |
| | PG-Mask-DS | 44.5 | 22.5 | 44.1 | 19.0 | 43.7 | 16.6 | 84.7 | 69.2 | 84.6 | 57.7 |
| | PG-Mask-BN | 55.0 | 32.3 | 54.6 | 26.0 | 54.0 | 19.8 | 84.5 | 63.8 | 83.9 | 47.3 |
| Certified Detection | MR | computationally infeasible | | | | | | 87.6 | 82.5 | 84.2 | 78.1 |
| | PatchGuard++ | 61.8 | 36.3 | 61.6 | 33.9 | 61.5 | 31.1 | 82.0 | 78.8 | 78.2 | 74.1 |
| | ScaleCert | 62.8 | **60.4** | 58.5 | **55.4** | 56.4 | **52.8** | 83.1 | 81.0 | 78.9 | 75.3 |

## 4.3 Detailed Analysis of ScaleCert

The insight of ScaleCert is to leverage the intrinsic differences of SINs in patched and benign images to achieve effective defenses. There are two key parameters affecting the clean and certified accuracy: the pruning rate in the shadow region and the occluding window size to target the adversarial patch (more discussion in Appendix A.2). In the ScaleCert algorithm, some hyperparameters can affect these two important parameters: the winner rate of SIN mask ($k$), the overlap ratio for merging the searching window ($\tau$), and the superficial layer selection ($l$). We analyze the impact of $k$, $\tau$, $l$ (in Appendix A.4) on the certified defense effectiveness and efficiency.

**Pruning Rate of SIN Mask:** We test the vanilla model accuracy, clean accuracy, and certified accuracy with different winner rate $k$ ranging from 10% to 20%, as shown in Table 3. The winner

rate refers to the top $k$ neurons in superficial layer and pruning others. The results show that: 1) the vanilla model accuracy maintains when the top-ranking rate is larger than 10%. 2) The selection of the winner rate $k$ is the tradeoff between the activation sparsification level and inference accuracy. On one hand, a smaller $k$ reduces more overhead of certified defenses by reducing the candidate sliding windows and filtering more noise prediction labels. On the other hand, it is more challenging to maintain the accuracy of the neural network with the heavy pruning activation map.

Table 3: Effect of Top-Ranking rate

| Top-Ranking Rate | 10% | | | 15% | | | 20% | | |
|---|---|---|---|---|---|---|---|---|---|
| Patch size | Model | Clean | Cert. | Model | Clean | Cert. | Model | Clean | Cert. |
| 1% | 73.6 | 61.4 | 56.0 | 74.2 | 60.5 | 58.4 | 75.0 | 62.8 | 60.4 |
| 2% | 73.8 | 58.7 | 51.8 | 74.6 | 58.7 | 54.9 | 74.4 | 55.9 | 53.3 |
| 3% | 73.2 | 54.0 | 48.6 | 74.3 | 57.3 | 49.6 | 74.2 | 56.5 | 51.6 |

**Occluding Window Sizes.** We also test the impact of overlap ratio threshold ($\tau$) when merging the occluding windows to reduce the searching windows for the better computing efficiency. When $\tau$ is larger, less windows will be merged, resulting with larger candidate searching window number and smaller occluding window size. Larger occluding window may hurt the model prediction accuracy. Less candidate searching window can accelerate the detection procedure and remove noises in the prediction map. Hence, the selection of overlap ratio is the tradeoff between occluding window number and window size. As shown in Table 4, the results indicate that ScaleCert can achieve both good certified accuracy and introduce the smallest computing overhead, when overlap ratio threshold is 0.3. We achieve over $100\times$ of speedup compared to MRD [17] and comparable computing overhead with PG-Mask-DS [25].

Table 4: Effect of overlap ratio threshold (PatchSize=2%, $k$=10%)

| Overlap Ratio ($\tau$) | 0.3 | 0.5 | 0.6 |
|---|---|---|---|
| Candidates | 25 | 36 | 64 |
| Execution Latency (V100 GPU) | 219$ms$ | 306$ms$ | 508$ms$ |
| Occluding Window Size | 77 | 64 | 58 |
| Clean Accuracy | 62.5 | 59.8 | 61.1 |
| Certified Accuracy | 56.6 | 55.2 | 56.0 |

## 4.4 Empirical recovery of ScaleCert

We evaluate the empirical recovery ability of our method on 10,000 randomly selected images that can be correctly by ResNet50 model from ImageNet. Firstly, for each image, a square patch with 5% pixels is generated and placed on the image at a random location. The classification accuracy drops to 14.0% after attack patchs. Then, we localize the patch of each image using algorithm 1. We tear the localized patch areas off from each image and put the masked image into inference to get the recovery label. The classification accuracy is significantly improved to 85.8%, which empirically shows that ScaleCert can effectively recover the true label from patch attacks by occluding the patch regions.

## 5 Conclusion and Future Work

In this paper, we propose the certified defense methodology ScaleCert for the detection of adversarial patches in high resolution images. We observe that adversarial patch intends to leverage localized superficial important neurons (SIN) to manipulate the prediction results. Hence, we propose SIN-based sparse techniques to reduce the adversarial region searching overhead and filter the prediction noises, which significantly improves the certified accuracy with good computing efficiency.

## Acknowledgements

This work is partially supported by the Beijing Natural Science Foundation (JQ18013), the NSF of China(under Grants 61925208, 62002338, U19B2019), Beijing Academy of Artificial Intelligence (BAAI) and Beijing Nova Program of Science and Technology (Z191100001119093) , CAS Project for Young Scientists in Basic Research(YSBR-029), Youth Innovation Promotion Association CAS and Xplore Prize.

## Broader Impact

As the widely usage of deep learning systems in the real world, our method can practically improve the robustness of DNN applications against adversarial patch attacks. Overall, we believe our paper has positive impact in the society but may potentially be misused to identify the weakness of DNNs.

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
