# A  Appendix

## A.1  Experimental Implementation Details

In Section 3.1, we empirically analyze the differences between SINs of patched and benign images. The benign image set consists of 10000 images randomly selected from ImageNet validation set. The patched images are generated by applying the adversarial patch to the random location of the images in benign image set. The adversarial patch, $\hat{P}$, is generated by maximizing the expectation of possibility for ResNet50 to output targeted malicious label $y_p$ (label 859, toaster in the evaluation) with all adversarial inputs $x_p$ derived from these 10000 images $\mathcal{X}$.

$$\hat{P} = arg \max_p \mathbb{E}_{\mathcal{X}}[logPr(\mathcal{M}(x_p) = y_p|x_p)] \tag{3}$$

**SIN distribution analysis.** We analyze the distribution of SINs in terms of both the standard deviation distance and cluster number. We first cluster the Top-200 neurons in the first layer with the MeanShift clustering algorithm and obtain the cluster number and the coordinates of cluster central nodes in the benign and patched images. Then we calculate the standard deviation distance with the following method: As formulated in Eq. (4), for each cluster $c$, we obtain the coordinate of central point $(\overline{x}_c, \overline{y}_c)$ by averaging the coordinates of SINs in cluster $c$. Then, we calculate the standard deviation distance $s_c$ as Eq. (5). Figure 1(b) illustrates the $s_c$ distribution of benign image set and patched image set.

$$\overline{x}_c = \frac{1}{N}\sum_{i=1}^{N} x_i, \ \ \overline{y}_c = \frac{1}{N}\sum_{i=1}^{N} y_i \tag{4}$$

$$s_c = \sqrt{\frac{1}{N}\sum_{i=1}^{N}[(x_i - \overline{x})^2 + (y_i - \overline{y})^2]} \tag{5}$$

**Prediction stability analysis.** In the prediction stability analysis (Figure 1(d)), we only consider the benign images (out of the 10000 selected images) that can be predicted correctly to avoid the interference of other factors.

**Adaptive attack methodology.** In Figure 1(e), we evaluate the adversarial patch detection rate of SIN-based detection methodology under adaptive attack. In this evaluation, the adversarial detection methodology is based on sliding windows to find the patch candidate region where the SIN ratio in it exceeds the threshold of 90%. Under adaptive attack, the adversary gets known the full knowledge of the defensive strategies. To steer clear of the localized SIN-based detection, the adversary trains the adversarial patch with the following optimization function by taking the superficial activation value into consideration. Compared to Equation (3), a penalty loss of activation value is considered. In this way, the adversary aims to build the adversarial patch with both good poisoning effects, but also tries to escape from the adversary candidate detection by reducing the superficial activation value of the patch region. $\alpha$ is the parameter to control the scale of the penalty loss in superficial activation value, $w_0$ is the weight of the first layer. $p$ is updated during the optimization of the loss function.

$$loss = -logPr(\mathcal{M}(x_p) = y_p) + \alpha * \sum(w_0 * x_p) \tag{6}$$

## A.2  Key Insight of ScaleCert

The insights of ScaleCert are shown in Figure 5. Figure 5(a) illustrates the superficial neuron importance of the first layer in one patched and benign image. For both patched and benign images, the nodes falling out of the Top-k rate are not essential for the prediction results and are pruned for computing efficiency and less prediction noise. SIN-based pruning is compatible with well-studied activation-based neural network compression techniques that are proved to introduce negligible accuracy loss to the DNN models. The distribution of retained SINs is distinct for patched images and benign images (as shown in Figure 5(b) and Figure 5(c) ). The basic idea of ScaleCert is based on the prediction stability after removing localized SINs in benign images.

Two factors are affecting the certified accuracy of ScaleCert: 1) Top-Ranking rate: it is the trade-off between the computing efficiency and model accuracy. 2) Occluding window size. The lower bound

of the window size is patch size. The upper bound of window size is determined by the trade-off between computing efficiency and certified accuracy. Therefore, we evaluate the clean and certified accuracy under different parameters in Section 4.3.

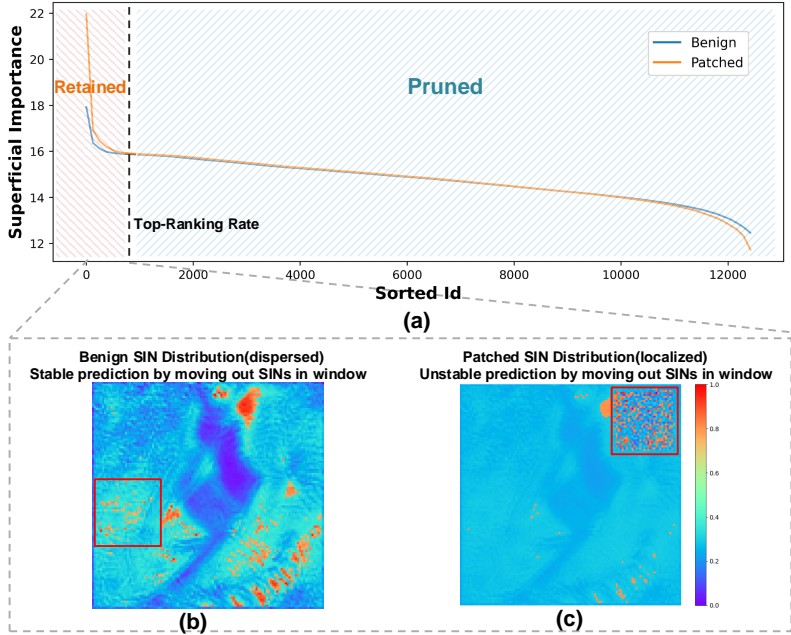

Figure 5: Key idea of ScaleCert. The sorted superfical importance of patched image is much larger than benign image on the retained top-k part while similar on the rest pruned part(a). Benign image has dispersed SINs and stable prediction when moving out those SINs(b).On the contrast, patched image has localized SINs and is sensitive to the occluding(c).

## A.3 DIN vs. SIN in Adversarial Patch Defenses

**DIN vs. SIN**. Deep neuron importance has been widely used for abnormal adversarial example detection in previous studies [6, 16, 29, 30]. Specifically, the class activation map (CAM) is a commonly-used technique to indicate the discriminative image regions to identify a particular class [34]. It leverages the weight of deep features (activation in the last convolutional layer) to indicate the importance of deep features for specific classes. We envision that deep feature importance is not a good candidate for adversarial patch detection because it cannot leverage the unique purturbation localization restriction of adversarial patches: 1) Both the benign images and the adversarial images exhibit the localized discriminative regions. Therefore, the discrimination of deep features in the benign images and adversarial images is not intuitive as that in superficial features (as shown in Figure 5(a, b, c)). It is challenging to eliminate the adversarial effect in deep features since the adversarial feature and benign feature are accumulated together. 2) Calculating such deep feature importance is time-consuming, which demands the gradient information and the complete backward propagation process.

We propose the superficial input feature importance as the metric for discrimination analysis based on the intuition that in order to efficiently manage the output prediction results with a very small region of the input data, the adversarial patch must incur large activation from the first place instead of the accumulation of the deep feature extraction.

**PatchGuard vs. ScaleCert.** To defend against adversarial patch attacks, PatchGuard leverages the deep feature importance analysis and clips the potential malicious deep features (with large weight values) based on robust aggregation. To isolate the adversarial features from benign features, PatchGuard tries to restrict the adversarial effect based on small reception field techniques. However, a smaller reception field may introduce large accuracy degradation (BagNet introduces about 20% of accuracy degradation compared to ResNet [25]). Additionally, for deep neural networks with more layers, small kernel sizes still result in a large reception field and raise the difficulty to isolate and distinguish malicious features from benign features. ScaleCert, on the other hand, optimizes the

defense techniques based on SIN-based neural network sparsity, which utilizes the localization bound restriction of patch attacks and introduces negligible loss to model accuracy [11].

## A.4 Sensitivity Study of ScaleCert

**Large patch defenses.** We also test the certified accuracy for the cases with super large adversarial patches, as shown in Table 5 and Table 6. With the increasing of patch size, the certified accuracies of all the defenses decrease. However, for ImageNet dataset, ScaleCert still retains the certified accuracy above 40% even under extreme cases that the patch contains 8% pixels of the input images.

Table 5: Certified accuracy comparison with large patch sizes

| Dataset | | ImageNet | | | | Cifar | | | |
|---|---|---|---|---|---|---|---|---|---|
| Patch size | | 5% pixels | | 8% pixels | | 5% pixels | | 8% pixels | |
| Accuracy | | clean | robust | clean | robust | clean | robust | clean | robust |
| Certified Recovery | IBP | computationally infeasible | | | | 24.8 | 17.8 | 19.0 | 13.8 |
| | CBN | 53.5 | 1.4 | 53.5 | 0.5 | 83.2 | 0.8 | 83.2 | 0.1 |
| | DS | 44.4 | 11.7 | 44.4 | 8.8 | 83.9 | 46.3 | 83.9 | 34.8 |
| | PG-Mask-DS | 43.1 | 13.2 | 42.2 | 9.5 | 84.7 | 47.6 | 84.3 | 35.5 |
| | PG-Mask-BN | 52.8 | 9.2 | 52.0 | 5.4 | 83.4 | 26.8 | 82.6 | 16.9 |
| Certified Detection | MR | computationally infeasible | | | | 82.1 | 75.4 | 79.9 | 72.0 |
| | PatchGuard++ | 61.4 | 25.3 | 61.5 | 22.2 | 73.5 | 67.6 | 70.8 | 64.2 |
| | ScaleCert | 54.0 | **48.7** | 50.6 | **42.7** | 76.3 | 71.2 | 73.5 | 67.6 |

Table 6: Certified accuracy with different Top-Ranking rate

| Top-Ranking Rate | 10% | | | 15% | | | 20% | | |
|---|---|---|---|---|---|---|---|---|---|
| Patch size | Model | Clean | Cert. | Model | Clean | Cert. | Model | Clean | Cert. |
| 5% | 72.9 | 53.0 | 44.4 | 74.2 | 51.3 | 47.6 | 74.6 | 54.0 | 48.7 |
| 8% | 72.7 | 48.4 | 40.2 | 73.3 | 46.0 | 41.6 | 73.8 | 50.6 | 42.7 |

**Superficial layer selection.** Additionally, we test the impact of using different superficial layers for the SIN mask computing and the results are shown in Table 7. When the superficial layer is closer to the input, the targeted searching region is more precise and smaller. Otherwise, the searching region is larger due to the reception field effects. The performance drop of Layer 2 is introduced by the MaxPooling layer between Layer1 and Layer 2 (ResNet50) which leads to information losses and pruning in Layer 2 would hurt the performance. We suggest that do not select the superficial layer right after the MaxPooling layers and recommend using the first superficial layer because of both good accuracy and less computing overhead.

Table 7: Effect of Superficial Layer Selection on Certified Accuracy

| Superficial Layer | 1 | 2 | 3 |
|---|---|---|---|
| Model Accuracy | 73.8 | 69.8 | 73.4 |
| Clean Accuracy | 58.7 | 47.0 | 54.0 |
| Certified Accuracy | 51.8 | 42.4 | 49.4 |