# OpenReview forum: "ScaleCert: Scalable Certified Defense against Adversarial Patches with Sparse Superficial Layers"
_NeurIPS.cc/2021/Conference — NeurIPS 2021 Poster_

### Official Review · Reviewer_sS3e · 2021-07-04

**Rating:** 6
**Confidence:** 5

**Summary:**

This paper proposed an algorithm called ScaleCert that attempted to improve the accuracy of certified detection against large-resolution adversarial samples. The experimental analysis about the motivation in the paper is great, the logic is clear and easy to understand, and the experiments prove the efficiency (time consuming) of the method.

**Limitations And Societal Impact:**

They have mentioned the limitations in the Appendix; the certified accuracy of ScaleCert maybe a little bit worse than the original accuracy. I don't see that is a limitiation to me.

**Main Review:**

1.	Although the related work mentions attack recovery and robust detection, the title as well as the abstract make this work appear not to be doing adversarial detection and may mislead researchers in the field of adversarial machine learning.
2.	Although the authors' analysis is better, I still have doubts about whether ScaleCert is provable. On the one hand, some adversarial patches can make the activation values of the network concentrated in the wrong places, such as [1], and it is possible to circumvent detection; on the other hand, there is no guarantee that the pruned network can always classify the images correctly after masking different locations, that is, the adversary can carefully design patches to lower the activation values to circumvent detection, but make the non-patched regions wrongly classified after being masked. And according to the algorithm, the patch may be detected, but causes a false alarm, and such a patch is harmful to the subsequent processing. Summing up the above two points I think ScalCert is a detection with high security for general attacks, but not provable.
3.	Why is the clear ACC of 2% size patch lower than the 3% case in the ImageNet experiment in Table 2? Detection accuracy should not be compared to the accuracy of robust defense, which are two completely different tasks. Experiments should give AUC values and not just detection accuracy. These experiments make me unconvinced of the validity of the method.


[1] Subramanya, Akshayvarun, Vipin Pillai, and Hamed Pirsiavash. 2019. "Fooling network interpretation in image classification." In Proceedings of the IEEE International Conference on Computer Vision, 2020-29.


**Time Spent Reviewing:**

8

---

> ### Author Response · Authors · 2021-08-09
> **Response to Reviewer sS3e**
>
> Dear review sS3e:
>
> We are deeply sorry for the misunderstanding on the provableness of our method. We made a more clean illustration and we are certainly confident that our SacleCert will lead to a lower bound of the optimal certified accuracy (please refer to the Q2), and we also address your concerns on the experiments in the rebuttal. Please feel free to let us know if you have additional comments and we are always happy to discuss them here.
>
> **[Q1] Certified detection v.s. certified recovery:**
>
> We will rephrase the abstract and introduction to emphasize the difference between certified detection and certified recovery. And please also refer to the **[General Response]** above that the certified detection is also empirically useful and is beneficial to the prediction accuracy and friendly to large scale datasets.
>
> **[Q2] Provable of our method:**
>
> Thanks for your comments. Our certification algorithm provides a lower bound of optimal certified accuracy (Line 273-276). Refer to the definition of our certification problem in Theorem 1 (Line 152), we try to certify whether the input image is clean or safe w.r.t. **a specific label $y$** under attack.  If the nature output is not $y$ then the certification is False. This definition is borrowed from mature image classification certifications and is also widely accepted in recent literature such as [PatchGuard and PatchGuard++].  If the certification returns True then the input is definitely safe, otherwise, it may be a false alarm or patch attacked.
>
> The two cases you mentioned fall into the false alarms in our algorithm and do not hurt our certified proof.
> - Case1: "there is no guarantee that the pruned network can always classify the images correctly after masking different locations": In terms of certified proof, we have no need to guarantee such a case under the certification problem. Our method either can classify correctly or yield a false alarm. The certified and clean accuracy (already excluding the false alarm) in the experimental evaluation indicates that our methodology can work well and greatly surpass the related work.
> - Case2: "Interpretation attack that makes the activation values of the network concentrated in the wrong places" Our method treats it with no difference from nature input and assures the certified proof:
>   - If the adversarial activations (affected by the adversarial patches) are not pruned, we will test the prediction results by masking them out.
>     - If all the occluding predictions have the same label, the attack fails.
>     - if the occluding predictions are not consistent, we raise the alert (possibly it is a false alarm but we will not miss any effective adversarial patches).
>   - if the adversarial activations (affected by the adversarial patches) are pruned, no adversarial effects will be retained to affect the prediction results.
>
> In conclusion, when the adversarial patch size is smaller than our filter size, our method is attacking agnostic and can always localize the patch as long as all masked out inputs can be classified correctly for the original clean input.
>
> Thanks for your comments again, It's really helpful to us to refine our paper and clarify the confusing part about the definition of the certification.
>
>
> **[Q3] Why is the clean ACC of 2% size patch lower than the 3% case in the ImageNet experiment in Table 2:**
>
> ​In the current version of the paper, the random seed is not fixed so that involved some randomness. We rerun the experiments by using different seeds and report the average results of 5 trials as follows:
>
> | Patch size  |  Model Acc |     Clean Acc   |      Cert Acc       |
> |:---------------:|:--------------:|:--------------------:|:--------------------:|
> |     2%          |      75.0     |          58.2          |      54.9             |
> |     3%          |      74.9     |          56.5          |      52.3             |
>
> As we can see the clean accuracy of patch size = 2% is higher than patch size = 3%.
>
> **[Q4] Experiments should give AUC values and not just detection accuracy:**
>
> AUC is not suitable for our experiments. We focus on classification tasks rather than object detection tasks in this paper.  Although our ScaleCert works in a detection manner, there is no concept of the confidence score/threshold to calculate AUC because the certification system is guaranteed to **not** provide false-negative results.

---

> > ### Comment · Reviewer_sS3e · 2021-08-21
> > **Concerns about Algorithm 2**
> >
> > *Comparison in experiments*
> > - Patch localization extended by provable detection can be used to help with empirical recovery defenses but is not appropriate compared to certified recovery defenses. It adds to the misleading nature.
> > *Randomness in experiments*
> > - Since errors due to randomness in experiments can be misleading, you must report the error bar.
> > *Questions about Algorithm 2*
> > - I have some questions about Algorithm 2, if I misunderstand this please clarify it. Algorithm 2 relies on the assumption that the prediction when the patch is masked is not consistent with the original prediction. Is this assumption proven? If a part of the image will be classified from $c_1$ to $c_2$ when it is masked, and the adversary places the patch on this part and directs the attack to $c_2$, then Algorithm 2 will output True.

---

> > > ### Author Response · Authors · 2021-08-26
> > > **Response to concerns about Algorithm 2**
> > >
> > > Dear Reviewer sS3e,
> > >
> > > Thank you so much for your encouraging feedback and nice suggestion. We list the response below to address your concerns about Algorithm 2 and randomness in our experiments.
> > >
> > > **[Q1] Patch localization extended by provable detection can be used to help with empirical recovery defenses but is not appropriate compared to certified recovery defenses. It adds to the misleading nature.**
> > >
> > > Thanks for your comment. We clarified this part in general responses above and we will make it clearer in the revision.
> > >
> > >
> > > **[Q2] Since errors due to randomness in experiments can be misleading, you must report the error bar**
> > >
> > > The randomness of our experiments comes from the masking-out location during fine-tuning the classification model. To eliminate the concern about randomness, we have trained more models with different seeds, but remain other settings unchanged.  The table below shows the *mean* and *variance* of 10 trials with different random seeds.  The results of the table indicate that the consistent phenomenon that clean accuracy and certified accuracy decrease with the increase of patch size.
> > >
> > > | Patch size | Clean acc mean  | Clean acc var   |  Cert acc mean   |    Cert acc var  |
> > > |:--------------:|:----------------------:|:--------------------:|:----------------------:|:--------------------:|
> > > |        1%      |         62.77          |          0.89         |          60.36         |         0.27          |
> > > |        2%      |         58.62          |          0.81         |          55.39         |         0.81          |
> > > |        3%      |         56.40          |          0.58         |          52.84         |         0.72          |
> > >
> > >
> > > **[Q3] Questions about Algorithm 2**
> > >
> > > Yes. Algorithm 2 relies on the assumption that prediction when the patch is masked is not consistent with the original prediction.
> > > When the prediction by the occluded patch is consistent with the original prediction, either patch failed to attack:
> > >   -  the original output label is wrong, it is the wrong prediction, not a successful attack
> > >   -  the original output label is true, this patch failed to attack
> > >
> > > or the input is benign.
> > >
> > > But the example you given is not totally correct: \
> > > $c_1$ is the original label without occluding. Algorithm 2 will output *False* since the occluding prediction is not consistent with the original label. It is true that it will be a false alarm when the part of the benign image will be classified from $c_1$ to $c_2$ without patch attack (it doesn't hurt the certification). Therefore, to reduce the false alarm rate, we train the model by *randomly masking subwindow* in the input (a similar technique used in MRD). We will also add the training details in the revision.

---

> > > > ### Comment · Reviewer_sS3e · 2021-09-01
> > > > **I decide to raise my score based on the rebuttal**
> > > >
> > > > Most of my concerns have been addressed. I will raise my score to 6.
> > > >
> > > > Two concerns.
> > > >
> > > > - I think the comparison of provable detection and provable recovery should not be placed in the main text. The comparison is meaningless due to the difference in settings.
> > > >
> > > > - The statement related to the algorithm is not very clear. I suggest that the authors clearly explain the algorithm in future revisions. In addition, Line 246-257 is copied from PatchGuard++, which is not allowed.

---

> > > > > ### Author Response · Authors · 2021-09-01
> > > > > **Response to concern about the comparison**
> > > > >
> > > > > We are glad to address your concerns. We appreciate your positive feedback.
> > > > >
> > > > > For the comparison with provable recovery methods, as we replied in the general response that our method can potentially do provable recovery because we can locate the patch locations. We will modify the comparison and make clarifications in the revision.
> > > > >
> > > > > Thanks again for your valuable comments and we will definitely modify the paper according to your constructive suggestions and adding the experimental results we obtained during rebuttal.

---

### Official Review · Reviewer_3JJX · 2021-07-15

**Rating:** 6
**Confidence:** 4

**Summary:**

This paper proposes to leverage superficial important neurons (SIN) to prune DNN to reduce the sensitivity of DNN to adversarial patches, thus resulting in certified defense of adversarial patches. The approach outperforms existing approaches largely on large ImageNet dataset.

**Limitations And Societal Impact:**

The authors adequately addressed the limitations.
The authors may need to include the discussion about the potential negative societal impact of their work if the manuscript is accepted.

**Main Review:**

Pros:
- The usage of SIN, or general interpretation/compression technique, for certified defense is quite novel.

- The empirical evidence that motivates the method seems compelling.

- Strong empirical results.

Cons:
- Method:
1. The proposed method, ScaleCert, is an attack detection method that does not aim to recover from the attack. The certification counts the input benign example as "certified" as long as the method outputs the correct label or raises an alert. Thus, in contrast to attack recovery methods, the proposed method is less useful in practice than what the certification result indicates since it may be too conservative for slightly patched inputs.

2. If I understand correctly, the SIN mask should be static given a model? Does it mean that the model only uses sparse, scattered, and tiny regions to sense the whole image space? If this is the case, how does the masked model preserve high benign accuracy given diverse key regions across different benign input images? Is the SIN mask generated only based on training dataset statistics?

3. The approach is limited to CNNs.

- Empirical result:
1. As shown in Table 2, the method is not as good as existing methods on low-resolution image datasets like CIFAR-10.

2. It would enhance the paper if there are more studies about the empirical patch detection ability, given that Line 14 in Algorithm 1 provides the ability to locate the patch.

- Writing: The details on SIN is lacking in this paper. For example, what is the formal definition of SIN? How does SIN determine what neurons to mask? How does the static SIN mask be generated in experiments?

Minor: Line 348, "sparese" => "sparse"

In summary, given the strong empirical results, interesting findings, and novel techniques, I weekly support the acceptance of the paper. If my concerns can be adequately addressed, especially those on SIN, I would consider raising my score.

**Time Spent Reviewing:**

3

---

> ### Author Response · Authors · 2021-08-09
> **Response to Reviewer 3JJX**
>
> Dear Reviewer 3JJX,
>
> We thank you for the encouraging feedback especially on the clarification on the SIN mask. We list our response below. Please don’t hesitate to let us know for any additional comments.
>
> **[Q1] ScaleCert is not aiming to recover patch attacks, being less useful in practice and too conservative for slightly patched inputs. It would enhance the paper if there are more studies about the empirical patch recovery ability:**
>
> Thanks for the great suggestion. Although certified detection relaxes the certification compared to certified recovery, certified detection is important since it is empirically easy to eliminate the patch effect (aka. recovery) by occluding the potential patch locations after the detection.
>
> We perform the experiments to evaluate the ability of patch recovery. Results show that tearing the localized patch areas off the images would improve the classification accuracy from 14.0% to 85.8%, which empirically shows that our method can effectively recover the true label from patch attacks by occluding the patch regions. Please refer to **[General Response]** above for more details.
>
> **[Q2] SIN mask is static given a model? Does it mean that the model only uses sparse, scattered, and tiny regions to sense the whole image space?**
>
> No. The SIN mask is dynamically generated by choosing the top-k activation neurons of the superficial important layer during the inference for each image. See Figure 2, Section 3.2 for more details.
>
> Due to the pruning operation, we only use top-k neurons(SINs) of the superficial important layer for whole image inference. However, the SINs can be localized and continuous around the object of the image but not always sparse and scattered for models are trained to locate and classify the object as shown in Figure1 (a).
>
> Besides, neural network pruning, aiming at reducing the size of a network by removing weights or neurons, has been well developed in recent years. By removing less important weights or neurons, the pruned models have fewer parameters while maintaining high accuracy or performance.
>
> **[Q3] The approach is limited to CNNs:**
>
> Yes. So far, ScaleCert is based on superficial important neurons that are supposed to appear in the patch-based attack process due to the localized feature aggregation of the shallow layer of CNN. But considering CNN is the most widely used architecture recently, it would not be a huge limitation of this work.
>
> **[Q4] The method is not as good as existing methods on low-resolution image datasets like CIFAR-10:**
>
> CIFAR-10 images are very small with the only size of 32x32, and thus are much more sensitive to pruning operations and masking operations. Therefore the certified accuracy drops slightly compared to MRD.
>
> However, ScaleCert greatly surpasses other studies in a larger-scale dataset like ImageNet that is more important and valuable for practical scenarios in real use.
>
> **[Q5] The details and formal definition of SIN? How does SIN determine what neurons to mask? How does the static SIN mask be generated in experiments?**
>
> **SIN**: Giving the feature map of the superficial important layer, we sort the neurons by the sum of their activation through channels and regard the top-k neurons as the superficial important neurons(SIN).  As described in Section 3.2, SIN is not static but dynamic for each image during their inference in experiments.
>
> **Formal definition of SIN**:  Suppose we use the layer $l$ as the superficial important layer, and the $ \\mathcal{N}^{l}_{i} $ is notated as the feature map of $i$-th channel of layer $l$. The SIN $ n^{l} $ is defined as follows.
>
> $ n^{l} = \\{ Top_{k}( \\mathcal{N}^{l}) \\} \\quad where \\quad \\mathcal{N}^{l} = \\sum_{i=1}^{c} \\mathcal{N}_{i}^{l}   $
>
> **SIN-mask**: SIN-mask is not static, but dynamically generated according to SINs when taking in different inputs. Specifically, SIN-mask is 2 dimensional and is the same size as the feature map of the superficial important layer. Giving the feature map of the superficial important layer, SIN-mask is the mask where the positions of SINs are set to 1 and others are set to 0.

---

> > ### Comment · Reviewer_3JJX · 2021-08-29
> > **Response to Authors' Comment**
> >
> > Thanks to the authors for the detailed response and additional experiment results. Most of my concerns are addressed.
> >
> > **A follow-up question**:
> >
> > I guess masking any region would lead to ignorance of that region during $\mathcal{M}$'s inference since the region will not contain top-k SIN neurons due to zero activation. Is this true? If it is true, $\mathcal{M}$ cannot apply pixel color standardization during data preprocessing and even cannot add bias terms which is a bit weird. if it is not true, in
> > https://1drv.ms/b/s!Au_82WgbLBo4bWD3Y3n4D0QSNfM?e=KRuWem the bottom right subfigure may still choose the patch region as top-k neurons and lead to $y'=c_2$ which falsifies the certification.
> >
> > Besides this follow-up question, the paper certainly needs more polishing since the current version is hard to understand due to many missing details. It may be difficult to truly understand the work without reading the authors' responses for all reviewers. Besides those promised by authors, I would suggest the authors making these changes in revision:
> >
> > - Clearly define SINs and states the important properties of SINs, e.g., what is the SIN of the masked region?
> >
> > - Clearly and formally define the inference protocol of $\mathcal{M}$. It seems to be first computing SINs then dynamically pruning the network to only pass the signals for top-k SIN neurons. The final inference protocol that takes windows into account runs multiple $\mathcal{M}$ inferences independently --- the top-k SINs are independently computed for each masked input.
> >
> > - A detailed proof of Theorem 1 with more illustrations.
> >
> > - Training details for reproducibility. The current submission lacks many training details nor contains code implementation. Please make sure to include them especially the description of training details.
> >
> > - Compress Section 3.1 to gain space.
> >
> > Overall,
> > I think the work itself would definitely be worth presentation at top-tier venues like NeurIPS, given its strong empirical result, novelty, and solid technique. However, the work is not presented well with these missing details, which prevents me from increasing the score.

---

> > > ### Author Response · Authors · 2021-08-31
> > > **Response to follow-up question**
> > >
> > > Thanks for your reply. We appreciate your encouraging feedback.
> > >
> > > For data preprocessing, we can support it by applying a little modification to our method:
> > >
> > > The Conv1 layer can be regarded as a linear operation for each input channel, we change it to $ a =[W]_{+} x + b $  that clips the weights by 0. Consider our model always performs pruning, the smaller outputs of layer 1 are negligible so this change will not hurt the accuracy (We test it in Cifar with Resnet18 model and the test accuracy changes from 94.65% to 94.39% after clipping the weight of Conv1 layer).
> > >
> > > In that way,  data preprocessing methods such as color standardization or adding bias, will not change the order of the pixels, which means they do not affect the order of the activation values since the weights are always greater or equal to zero. Therefore, with the constraint on weight, we can use those data preprocessing methods commonly while maintaining the correctness of the certification of our algorithm.
> > >
> > > For other suggestions on clarification and reorganization of our current version, we definitely will fix them in the final version.  And we will release our code in the revision.

---

> > > > ### Comment · Reviewer_3JJX · 2021-08-31
> > > > **Thanks for Response to Follow-up Question**
> > > >
> > > > I thank the authors for responding to the follow-up question.
> > > >
> > > > This response addressed the follow-up question.
> > > >
> > > > Though there is a minor limitation for applying weight cropping: the method may be challenging to generalize to selecting deep Conv layer for computing SIN. Because we will need to crop the weights from the selected layer all the way back to the first layer which may deteriorate the performance much. Therefore, we have to hope that shallow Conv layers do a good job selecting key regions. The current test accuracy may indicate that this is true to some extent, but there is still a gap between the current test accuracy and the common accuracy on ImageNet (typically >=70%) and this limitation may prevent the further improvement of the method to achieve that level.
> > > >
> > > > Anyway, this minor limitation mainly impedes future improvements and may not count much when evaluating the current work. So I will keep my score unchanged.

---

### Official Review · Reviewer_EwCy · 2021-07-17

**Rating:** 6
**Confidence:** 4

**Summary:**

In this paper, the authors propose a scalable certified defense framework to detect patch attacks. The authors empirically show that patch attacks rely on localized Superficial Important Neurons (SINs) to cause misclassification. By occluding image patches corresponding to these localized neurons, the predictions on patched images become unstable, while those of benign images rarely change. The proposed approach ScaleCert uses these SINs to check if a patch attack is present by occluding certain candidate regions and observing if the prediction changes. The proposed method leads to significant improvements in certified detection accuracy on ImageNet, while having lower computational cost, thus promoting its usage in the real world.

**Limitations And Societal Impact:**

No

Limitations: The proposed method detects the presence of a patch, and does not recover the true label.


**Main Review:**

Strengths:
 -  While many existing certified detection methods (MRD, PatchGuard++) are based on the idea of occluding patches in the input image, and observing the stability of the output prediction, the key insight in this work is in the SIN-based pruning step, where top-k neurons are retained and the rest are pruned. This improves the scalability of the approach and also leads to less noise in prediction labels, improving the certified accuracy significantly at a low computational cost
 -  The results show huge improvements in certified detection accuracy on ImageNet, achieving 60.6% against 1%-pixel patches, significantly surpassing the highest record of 36.3% in current state-of-the-art methods. The work also doesn’t incur a huge computational cost compared to other defenses, which is beneficial for real-world adoption.
 -  The paper is clearly written and well organized. It provides enough information for the reader to reproduce its results.

Questions for rebuttal/ suggestions for improvement:
 -  Could the authors share the execution latency on Imagenet for PatchGuard++ as well, which can be compared with the results in Table-4?
 -  Could the authors clarify how the top-k neurons are found for a given layer? Is it an average/ max across all activations at that layer?
 -  The process of certification and inference seem to be quite similar. For a clean input image, if occlusion prediction gives consistent outputs and if the prediction is same as the true label, it is said to be certified. The same process is also used during inference of a clean image for prediction. So does this mean that the difference between clean and certified accuracy is from images whose predictions are consistent and incorrect? It would be worth visualizing such images to see if they are confusing visually as well.
 -  Adding more detailed captions to tables and figures would improve the readability. The x-axis label for Fig.1(e) is missing.
 -  It could be made more clear throughout the paper that this is a certified detection method rather than a certified prediction/recovery method, perhaps in the title as well. It seems unfair to directly compare certified accuracy of recovery methods to detection methods as it’s easier to detect whether a patch is present than it is to predict the correct class in any patched image. A note on why this method cannot be used for certified prediction would also help the reader understand the same better.

Limitations/ Possible concerns:
 -  As seen in Table-5 of the Supplementary, clean accuracy drops significantly for 5% and 8% pixel patch attacks.
 -  The process of selecting only top-k neurons for inference could potentially make the network more vulnerable to minor distribution shifts such as addition of gaussian noise/ other corruptions when compared to a normally trained model.


**Update Post-Rebuttal**

I thank the authors for their detailed replies to clarify all concerns. Although the work is novel and shows a large improvement over prior work, there are many unclear parts and missing details, specifically about the certification, due to which I lower my score to 6.


**Time Spent Reviewing:**

10

---

> ### Author Response · Authors · 2021-08-09
> **Response to Reviewer EwCy**
>
> Dear Reviewer EwCy,
>
> We would like to thank you again for recognizing our main contributions and providing very constructive reviews. We have added the additional experiments as you suggested, and also will revise the paper following your suggestions.
>
> **[Q1] Execution latency on ImageNet for PatchGuard++ :**
>
> The execution latency is about 40ms/image. Our method has higher latency than PatchGuard++ due to the multiple complete inferences with different occluding windows. However, there are many redundant feature computations in this process and we will keep the performance optimization as the future work.
>
> Although the execution latency of PatchGuard++ is less, its certified accuracy is much lower than ours (36.3% vs. 60.6%). Moreover, the usage of PatchGuard++ is restricted to the models with small receptive fields, while our method is more general with fewer restrictions on the targeted models.
>
> **[Q2] How the top-k neurons are found for a given layer?**
>
> Given the superficial important layer,  we sum each neuron of the layer across all channels and regard the activation values as their importance. Then, we sort the activation values to get the top-k neurons. Line 214-215 explains the process.
>
> **[Q3] The difference between certification and inference:**
> - Inference:
> For an input image, It is classified correctly if occlusion prediction gives consistent outputs as the true label $\textbf{or}$ the majority voting results are the true label.
> - Certification:
> For an input image, $\textbf{if and only if}$ the occlusion prediction gives consistent outputs as the true label, it is certified-detection. We will visualize such images in the revision.
>
> **[Q4] Writing issues:**
>
> We will add more detailed captions to tables and figures to improve the readability.
>
> **[Q5] It seems unfair to directly compare certified accuracy of recovery methods to detection methods as it’s easier to detect whether a patch is present than it is to predict the correct class in any patched image. A note on why this method cannot be used for certified prediction would also help the reader understand the same better:**
>
> Thanks for the suggestions. We will clarify in the further step that it is a certified detection study to avoid confusion. During the comparison (in Table 2 and 5), we separate detection methods and recovery methods into two categories and ScaleCert is mainly compared with detection methods(MR, PatchGuard++). Please refer to **[General Response]** above for more details.
>
> **[Q6] Clean accuracy drops significantly for 5% and 8% pixel patch attacks:**
>
> With extremely large patches, large windows for masking will be used in our method, which may hurt the clean accuracy. However, under 8% pixel patch attacks, our method still achieves 42.7% certified accuracy, which is significantly better than other methods (0.5%-22.2%).
>
> **[Q7] The selection of top-k neurons seems to be more vulnerable to minor distribution shifts such as the addition of Gaussian noise/other corruption?**
>
> The effect of pruning on model robustness in terms of noise or corruption is still an open issue. Some studies show that network compression techniques have positive impacts on the model robustness [1][2][3].
>
> We evaluate the Gaussian noise effect on the pruned model. Specifically, we compare the model accuracy of original ResNet50 and pruned when taking inputs with Gaussian noises, as shown in the table below. The image tensors are normalized before inference. For the Gaussian noises, the mean is set to 0.1 and variance ranges in 0.01, 0.1, 0.5, 1. The result shows the trend that the pruned model performs better with the increase of noise variance. Overall, it is not obvious that the top-k pruned methodology is more vulnerable. We will keep research on this direction in future work.
>
> | Mean  |  Variance |  Original model  | 10%Pruned model  |  Acc difference  |
> |:--------:|:-------------:|:---------------------:|:---------------------------:|:--------------------:|
> |   0.1    |     0.01     |          73.4          |              72.4             |          1.0           |
> |   0.1    |     0.1       |          63.1          |              61.9             |          1.2           |
> |   0.1    |     0.5       |          33.4          |              33.5             |          -0.1          |
> |   0.1    |     1          |          13.2          |              15.9             |          -2.7          |
>
> [1] Diffenderfer, James, et al. "A Winning Hand: Compressing Deep Networks Can Improve Out-Of-Distribution Robustness." arXiv preprint arXiv:2106.09129 (2021).
>
> [2] Guo, Yiwen, et al. "Sparse DNNs with Improved Adversarial Robustness." Advances in Neural Information Processing Systems 31 (2018): 242-251.
>
> [3] Adnan Siraj Rakin, Zhezhi He, Li Yang, Yanzhi Wang, Liqiang Wang, Deliang Fan, “Robust Sparse Regularization: Simultaneously Optimizing Neural Network Robustness and Compactness”, 30th edition of the ACM Great Lakes Symposium on VLSI (GLSVLSI), September 7-9, 2020

---

> > ### Comment · Reviewer_EwCy · 2021-08-26
> > **Concern about the certification**
> >
> > I thank the authors for the detailed response.
> >
> > Could the authors clarify the condition in line-11 of Algorithm-1? Is an $r \times r$ patch of outputs that is different from $y^*$ the criteria for an alarm here? I guess the true label $y$ cannot appear in this algorithm which is meant for inference?
> >
> > What if an attack when occluded flips the output fewer times? (although I agree there would be at least $r \times r $ locations that occlude the patch or part of the patch completely.)
> >
> > Even for certified images (that pass Algorithm-2), it is possible to have regions that lead to inconsistent predictions when occluded. Such regions can exist in the part that does not correspond to the top-k activation regions of the clean image (I understand this may not happen in practice, but this is not provably ensured). Suppose occluding such a region flips the output from $c_1$ to $c_2$, and an attack with target $c_2$ is constructed near this region, this may mislead Algorithm-1. It is assumed that occluding the patch attack will flip the output from $c_2$ to $c_1$. However, since the nearby region is also occluded, the output may remain $c_2$ for most occlusions, leading to very few output label flips. This may violate the conditions in line 11 and 13 of Algorithm-1, causing misclassification although the image is certified.
> >
> > Could the authors clarify if this is possible?

---

> > > ### Author Response · Authors · 2021-08-28
> > > **Response to concern about the certification**
> > >
> > > We thank the reviewer's further questions about the certification. We clarify the misunderstandings as follows.
> > >
> > > **[Q1] The condition in line-11 of algorithm-1** \
> > > Yes, you are right.  An $r \times r$ patch of outputs that is different from $y^*$ is the criteria for an alarm here.
> > > Sorry for the notation abuse, the algorithm doesn't know the true label during the inference, here we mean to say there exist $y$ meet the condition not mean to the true label.
> > >
> > > **[Q2] What if an attack when occluded flips the output fewer times (condition in line-13)** \
> > > We want to emphasize that there would be at least $r \times r$ locations that occlude the patch completely (not part of the patch). Please see the definition of $r$ in line 203-204.
> > >
> > > For all the certified images $x$ that passed Algorithm 2, their generated patched images $\mathcal{A}(\mathbf{x})$  would satisfy the first condition in Algorithm 1 (line 11).  Because $r \times r$ regions would occlude the patch completely and the  masked-out images of $x$ would produce the consistent different label $y\prime$, the first condition (line 11) in Algorithm 1 would be satisfied for all the patched image $\mathcal{A}(\mathbf{x})$.  It won't go to the majority voting condition (line 13).
> > >
> > > **[Q3] The example suspected to violate the certification** \
> > > Thanks for the comments! We admit that the situation you constructed here is exactly the worst case in our certification system. But even in this way, our algorithm 1 will still trigger Line 11 and yield an alert. The key point is that our model always​ conducts Top-k dynamic pruning. So wherever the patch is, as long as the patch is occluded then the SIN area will be identical to the clean image and return the Ture label. In your case, the specific regions that lead $c_2$ will also be pruned when the patch is totally masked since there are out of Top-k in the clean image. We illustrate the procedure with all possible situations in Figure: \
> > > https://1drv.ms/b/s!Au_82WgbLBo4bWD3Y3n4D0QSNfM?e=KRuWem
> > >
> > > Sorry for the misleading, we will add more clarification on the significance of the sparsification because it's necessary and distinguished us from all existing methods.

---

> > > > ### Comment · Reviewer_EwCy · 2021-08-28
> > > > **Response to authors' clarification**
> > > >
> > > > I thank the authors for the clarification and appreciate the effort in illustrating all possible cases in a short time.
> > > >
> > > > I believe L254-255 of the main paper and Algorithm-2 (L-4) are not consistent, and this is the reason for the confusion. Algorithm-2 only ensures that sliding windows that overlap with $\mathcal{R}$ do not affect prediction results for image $x$ (Line-4 of Algorithm-2), whereas, L254-255 states the following:
> > > >
> > > > *The Algorithm 2 returns True indicates that all single sliding window $w_i \in \mathcal{w}$ do not affect the prediction results for image x.*
> > > >
> > > > If L254-255 was true, I agree that the certification is valid. However, I believe that imposing such a stringent constraint would bring down the certified accuracy a lot.
> > > >
> > > > If I assume Algorithm-2 is correct, the following in authors' response cannot be true always:
> > > >
> > > > *For all the certified images $x$ that passed Algorithm 2, their generated patched images $\mathcal{A}(\mathbf{x})$  would satisfy the first condition in Algorithm 1 (line 11).*
> > > >
> > > > This is because the Top-k region of clean image can be different from the Top-k region of the attacked image, and Algorithm-2 only guarantees consistent predictions while masking parts of $\mathcal{R}$ that correspond to top-k of the clean image (not top-k of attacked image).
> > > >
> > > > The Figure shared in the rebuttal also considers the top-k region of clean and attacked images to be the same, which need not be true.

---

> > > > > ### Author Response · Authors · 2021-08-28
> > > > > **Reponse to futher concern about the certification**
> > > > >
> > > > > Thanks for your quick reply! We are very glad to discuss with an expert like you to help us further improve and clarify our paper.
> > > > >
> > > > > We totally agree with you that the Top-k region of $x$ and $\mathcal{A}(\mathbf{x})$ are different when a powerful patch is outside the Top-k region of $x$. However, we believe there is a point that you may miss here: the sum area of Top-k regions for all  $\mathcal{A}(\mathbf{x})$ are always the same. And considering we use the first layer as the superficial important layer, so the most area that the patch can affect is just a little bit larger than the patch itself (caused by the stride and kernel size). So the Top-k region is not able to be changed by the patch arbitrarily. More specifically, the area affected by the patch will be exactly lost from the original Top-k region of $x$.
> > > > > Please see the Figure:
> > > > >  https://1drv.ms/b/s!Au_82WgbLBo4biPwUO_3mH_1kPk
> > > > >
> > > > > In Figure(b) (c) (d), we illustrate some possible cases that may be caused by the patch. Note that the green area lost from (a) can locate anywhere even not be continuous, but they are not able to locate outside the original Top-k region of $x$, since the pruned region has smaller activations.
> > > > >
> > > > > Based on that, as long as a window masked the patch, the  Top-k region will become Figure(a) and predicts $y^\prime$ as we showed in the previous figure:
> > > > > https://1drv.ms/b/s!Au_82WgbLBo4bWD3Y3n4D0QSNfM?e=KRuWem
> > > > >
> > > > >  We thank you again for your profound think of this work, and please let us know if you have any further questions.

---

> > > > > > ### Comment · Reviewer_EwCy · 2021-08-29
> > > > > > **Thank you for your response**
> > > > > >
> > > > > > I thank the authors for the clarification. This sufficiently addresses my concerns about the certification. I agree with the reviewer *3JJX* that a lot of important details/ explanations are missing in the current version of the paper. I encourage the authors to rephrase L254-255 in the paper and include more details about the certification (such as the figures and details shared in the rebuttal) in the final version.

---

> > > > > > > ### Author Response · Authors · 2021-08-29
> > > > > > > **Thank you for your support**
> > > > > > >
> > > > > > > We are glad to address your concerns, and we will definitely revise our paper that according to your suggestions.
> > > > > > >
> > > > > > > Thanks for your time and patience.

---

### Official Review · Reviewer_1feY · 2021-07-19

**Rating:** 6
**Confidence:** 4

**Summary:**

The proposed approach presents scalable certified defense against adversarial patch attack.  The method relies on pruning unimportant neurons and identifying regions in the input image which contain the adversarial patch. Once the patch location is identified, the label is predicted is using prediction consistency. A certification algorithm is provided which shows significant improvements over previous methods on ImageNet.

**Ethical Concerns:**

There are no ethical concerns with this paper.

**Limitations And Societal Impact:**

The authors have not addressed limitations and societal impact of their work. A discussion on this seems to be missing from the appendix.

**Main Review:**

Strengths:

— The method shows significant improvement compared to previous approaches on ImageNet, addressing an important problem of scalability for defenses against adversarial patch attacks.

 — The paper is well written and easy to follow.

— Detailed ablation study shows effect of individual component of the algorithm.

Weaknesses:

— Results are shown on only adversarial patch attack, which is one form of physical world attack. Other attack formulations such as Adversarial framing and Adversarial camouflage are not considered.

— Other large scale datasets such as Places can be considered for large scale datasets. ImageNet being an object centric dataset might be helpful in identifying the

patched region.

Additional Comments:

— Adversarial framing [1] and Adversarial camouflage[2] are more recent forms of physical world attack. Does the proposed approach defend well against such attacks?

— Although the proposed approach works well on ImageNet, more explanation is required for explaining the drop in performance w.r.t both clean and certified accuracy on Cifar10. This gap increases as the size of patch increases.

— The choice of superficial layer is an important experiment which does not lead to clear understanding of the method. Why does the performance drop for Layer 2 and increase again for Layer3 in Table 7 of the appendix?  A more thorough analysis seems necessary.

[1] -  arXiv:1812.04599

[2] - arXiv:2003.08757

-----POST REBUTTAL-----
The authors have addressed most of my concerns. Having gone through other reviews and the rebuttal, I agree that this is a good paper and I raise my score to 6.  I encourage the authors to include the discussion on the drop in performance for low resolution datasets and the guidelines for superficial layer selection.

**Time Spent Reviewing:**

8

---

> ### Author Response · Authors · 2021-08-09
> **Response to Reviewer 1feY**
>
> Dear Reviewer 1feY,
>
> We would like to thank you for your valuable comments. We are sorry for some misunderstandings in the current version of the paper. We addressed all of your concerns in the rebuttal and will rewrite a more comprehensive related work as you suggested.
>
> **[Q1] The effectiveness on adversarial framing and adversarial camouflage attacks:**
>
> Thanks for the references. The proposed defense methodology is generally effective on the adversarial attack with localized perturbations, such as adversarial patch and adversarial camouflage attacks. The evaluation results in Table 2 and Table 5 show the certified accuracy when the perturbations (either patch attack or camouflage attack) are within a bounding box of 1%~8% of the images.
> Adversarial framing attacks have different restrictions on the perturbation (in the frame of images). We will keep the advanced scenarios with different restrictions on the perturbation (adversarial framing and large camouflages) in future work.
>
> **[Q2] The concern of selecting ImageNet as the dataset (ImageNet is an object-centric dataset that might be helpful in identifying the patched region):**
>
> The ImageNet adopted in this work is at the high end of the benchmark scale evaluated by related studies (using MNIST, Cifar10, ImageNette, etc.). Moreover, the certified detection accuracy of our methodology is not correlated with the patch and object locations, which provides the certified proof for all the potential patch attacks (with different patterns and locations).
>
> **[Q3] Accuracy drop in CIFAR-10:**
>
> CIFAR-10 images are very small with the only size of 32x32, and thus are much more sensitive to pruning operations and masking operations. Therefore the certified accuracy drops slightly compared to MRD.
> However, ScaleCert greatly surpasses other studies in larger-scale datasets like ImageNet that is more important and valuable for practical scenarios in real use.
>
> **[Q4] Choice of superficial important layer:**
>
> Why Layer 2 performance drops: The performance drop of Layer 2 is introduced by the MaxPooling layer between Layer1 and Layer2 (ResNet50), which is commonly used for high-resolution datasets to reduce the number of parameters. The pooling layer leads to information losses and pruning in Layer2 would hurt the performance.
>
> Guidelines of superficial layer selection:
> - We suggest that do not select the superficial layer right after MaxPooling layers.
> - We recommend using the first superficial layer because of both good accuracy and less computing overhead.

---

### Author Response · Authors · 2021-08-08
**General Response (To Reviewer EwCy & 3JJX & sS3e)**

**Certified detection instead of certified recovery&The importance of certified detection:**

We will emphasize in the further step that it is a certified detection study to avoid confusion. During the comparison (in Table 2 and 5), we separate detection methods and recovery methods into two categories and ScaleCert is mainly compared with detection methods(MRD, PatchGuard++).

Although certified detection relaxes the certification compared to certified recovery, certified detection is important since it is empirically easy to eliminate the patch effect (recovery) by occluding the potential patch locations after the detection, which is important for practical scenarios.

Empirical recovery results: We evaluate the empirical recovery ability of our method on 10,000 randomly selected images that can be classified correctly by ResNet50 model from ImageNet. Firstly, for each image, a square patch with 5% pixels is generated and placed on the image at a random location. The classification accuracy drops to 14.0% after patch attacks. Then, we localize the patch of each image using algorithm 1. We tear the localized patch areas off from each image and put the masked image into inference to get the recovery label. The classification accuracy is significantly improved to 85.8%, which empirically shows that it can effectively recover the true label from patch attacks by occluding the patch regions.

---

### Decision · Program_Chairs · 2021-09-27

**Decision:**

Accept (Poster)

**Comment:**

The authors develop a novel scalable robustness certification technique to certify robustness of deep networks to adversarial patches. The paper provides a theoretically and empirically interesting contribution to the literature on robustness and verification of deep networks.

In the initial reviews, several concerns were raised around the correctness of the proposed approach. These were adequately addressed in the rebuttal. The paper would be acceptable for publication provided that the authors make the revisions suggested by the reviewers during the rebuttal phase in the camera ready version.

Specific detailed feedback from the reviewers on how to improve the paper:

1. L254-255 in the main paper needs to be rephrased to indicate that only sliding windows that overlap with  (corresponding to the benign image) do not affect prediction results for the image . Justification of why this is sufficient for certification needs to be included. This could include the figure shared by the authors in the rebuttal to illustrate that the top-k region of the attacked image cannot differ by more than a fixed amount from the top-k region of the original image. In addition, for the cases when the patch is occluded completely, the top-k region of the attacked image would be a subset of the top-k region of the benign image. This can be formally stated and proved.
2. It could be made more clear throughout the paper that this is a certified detection method rather than a certified prediction/ recovery method, perhaps in the title as well. It seems unfair to directly compare certified accuracy of recovery methods to detection methods as it’s easier to detect whether a patch is present than it is to predict the correct class in any patched image. A note on why this method cannot be used for certified prediction would also help the reader understand the same better.
3. Results on execution latency on ImageNet for PatchGuard++ (shared in the rebuttal) could be included
4. Details on how the top-k neurons are computed could be included
5. Adding detailed captions to tables and figures would improve the readability. The x-axis label for Fig.1(e) is missing
6. A discussion on false alarms and what is done to reduce this could be included.
7. Clearly define SINs and states the important properties of SINs, e.g., what is the SIN of the masked region?
8. Clearly and formally define the inference protocol. It seems to be first computing SINs then dynamically pruning the network to only pass the signals for top-k SIN neurons. The final inference protocol that takes windows into account runs multiple  inferences independently --- the top-k SINs are independently computed for each masked input.
9. A detailed proof of Theorem 1 with more illustrations.
10. Training details for reproducibility. The current submission lacks many training details nor contains code implementation. Please make sure to include them especially the description of training details.
11. Compress Section 3.1 to gain space.
12. Authors should include the discussion on the drop in performance for low-resolution datasets and the guidelines for superficial layer selection.
13. Comparison of provable detection and provable recovery should not be placed in the main text. The comparison is meaningless due to the difference in settings.
14. The statement related to the algorithm is not very clear. I suggest that the authors clearly explain the algorithm in future revisions. In addition, Line 246-257 is copied from PatchGuard++, which is not allowed.
15. Reporting error bars of results (shared in the rebuttal)
16. A discussion on limitations could be included: sub-optimal performance on low-resolution datasets, a large drop in clean accuracy with large patch attacks, and limited generalization (only works for image CNNs due to exploiting localized low-level features).